# Active School Transportation in Winter Conditions: Biking Together Is Warmer

**DOI:** 10.3390/ijerph16020234

**Published:** 2019-01-15

**Authors:** Anna-Karin Lindqvist, Marie Löf, Anna Ek, Stina Rutberg

**Affiliations:** 1Department of Health Sciences, Luleå University of Technology, 971 87 Luleå, Sweden; stina.rutberg@ltu.se; 2Department of Biosciences and Nutrition, Karolinska Institutet, 141 83 Huddinge, Sweden; marie.lof@ki.se (M.L.); anna.ek@ki.se (A.E.); 3Department of Medical and Health Sciences, Linköping University, 581 85 Linköping, Sweden

**Keywords:** active school transportation, children, empowerment, health promotion, parents, photovoice, physical activity

## Abstract

There has been a decline in children’s use of active school transportation (AST) while there is also limited research concerning AST in winter conditions. This study aimed to explore the prerequisites and experiences of schoolchildren and parents participating in an empowerment- and gamification-inspired intervention to promote students’ AST in winter conditions. *Methods:* Thirty-five students, who were aged 12–13 years, and 34 parents from the north of Sweden participated in the study. Data were collected using photovoice and open questions in a questionnaire and analyzed using qualitative content analysis. *Results:* The results show that involvement and togetherness motivated the students to use AST. In addition, during the project, the parents changed to have more positive attitudes towards their children’s use of AST. The students reported that using AST during wintertime is strenuous but rewarding and imparts a sense of pride. *Conclusion:* Interventions for increasing students’ AST in winter conditions should focus on the motivational aspects for both children and parents. For overcoming parental hesitation with regards to AST during winter, addressing their concerns and empowering the students are key factors. To increase the use of AST all year around, targeting the challenges perceived during the winter is especially beneficial.

## 1. Introduction

There is an association of walking or biking to school (i.e., active school transportation (AST)) with higher physical activity levels [1]. Even a modest increase in AST would result in a substantial increase in children meeting the public health recommendation of 60 min of daily physical activity [2]. Unfortunately, there has been a global decline in AST in recent decades and the proportion of children using AST varies greatly across countries [3]. For example, in the Netherlands, 79% of 12–17-year-old children use AST three or more days per week [4], compared to the United States (US), where less than 10% walk or bike to school [5]. The infrequent use of AST in the US suggests that additional work is required. When implementing interventional strategies, special attention should be paid to girls, minorities and those from lower socioeconomic groups [5]. The differences in AST between North America and Europe reflect their different policies on pedestrian safety and cultural norms in addition to the compact nature of European cities (that allows for short trips) [6]. In Sweden, the community and the built environment supports the promotion of physical activity and AST. However, only 57% of 6- to 15-year-old children use AST during spring and fall, with the prevalence found to be even lower in the winter (48%) [7]. AST has been found to increase with age, peak during adolescence and subsequently decline thereafter [8]. Moreover, the proportion of children engaged in AST is likely to further decline in the absence of interventions. In many countries, seasonal climate and weather-related variables need to be considered. The previously described seasonal difference found in Sweden was also recognized in a Finnish study, where cycling (unsurprisingly) was less common in winter, especially among girls and younger children [9]. These authors conclude that the potential for increasing children’s AST seems to be greatest in winter, especially among children living 1–5 km from school [9]. There is scarce research that has focused on seasonal differences. However, this is a pattern evident in other similar activities, such as children’s play, as it seems that even in a relatively mild climate, winter is considered to be the time to relax and stay indoors [10].

Based on a large body of evidence, health professionals and decision-makers are confident that the benefits from active transportation are worth pursuing [11]. However, there are gaps in the evidence concerning community-wide approaches, such as school programs aiming to equalize inequities in physical activity and sedentary behaviors [12]. Interventions to increase AST have been shown to be effective in evaluations conducted directly after the intervention, but there is little research on the sustainability of AST [13]. Involving children in AST discussions and taking the neighborhood setting into account might better inform the development of AST programs and enhance their sustainability [14]. Moreover, integrating children in the research process from the beginning is valuable and might shed a different light on the research questions [15]. Photovoice is a community-based participatory research tool where participants photograph things and the photos are used when discussing the issue. Moreover, photovoice enables the participants to express issues affecting their community through active engagement in research [16]. Participants are considered to be the experts and using photovoice will allow researchers to explore the perceived and experienced community environment where health behaviors are shaped and enacted [17]. Findings collected using photovoice might influence community policies or interventions aiming to address local needs and have been used in earlier studies to enable promotion of physical activity [18,19]. However, studies focusing on particular types of physical activity (e.g., walking and biking) and overlooked population groups (e.g., adolescents) are needed to identify environmental factors supporting physical activity in a community [20]. Therefore, this study aimed to explore the prerequisites and experiences of schoolchildren and parents participating in empowerment- and gamification-inspired intervention to promote children’s AST in winter conditions.

## 2. Materials and Methods

### 2.1. Design

To explore the prerequisites and experiences of participating in an intervention to promote AST in winter conditions, we have chosen a photovoice with a qualitative approach. The intervention, which is aimed at getting the students to use active transport and encouraging them to try cycling in winter, was created together with the end-users and built on the constructs of the Social Cognitive Theory [21]. The social cognitive theory provides a framework for understanding, predicting and changing human behavior and considers individuals within a collective context since people do not operate as isolated individuals. Moreover, it specifies a set of constructs, including knowledge, perceived self-efficacy, outcome expectations, goals, perceived facilitators and impediments to change. Self-efficacy is an important aspect of the Social Cognitive theory, signifying a fundamental belief in the individual’s ability to achieve a goal, and the degree of self-efficacy that an individual possesses directly affects his or her ability to change and ultimately affect health functioning [21]. Therefore, the project included activities to increase knowledge, enhance perceived self-efficacy among parents and students and utilize perceived facilitators and impediments to change. We have previously presented the theory underpinning this intervention [22].

### 2.2. Procedure and Participants

Information about the project was given to the principals of elementary schools in a municipality of northern Sweden. One school was chosen based on its interest in participating and its previous problems with heavy traffic during drop-off and pick-up hours. The principal and teachers representing two classes gave their informed consent to participate in the project. The students and parents were informed in person about the project by the authors. Thirty-five parents gave their written informed consent for their children to participate and all of those agreed to participate. The study was approved by the Regional Ethical Board in Umeå, Sweden (No. 2018-10-31M). Information about the municipality where the school is located and the participants is given in Table 1.

The intervention was initiated in phase 1, which used an empowerment approach. Students were asked to come up with ideas and elements to include in the project for motivation and ensuring a successful outcome. An overview of the timeline is shown in Table 2. The students highlighted prioritizing safety (winter tires and helmets), focusing on cooperation and mutual benefits, and stated that convincing the parents to let them participate would be an important determining factor of the success of the project. Furthermore, the students expressed a wish to get hot chocolate in the mornings and get a reward as a class activity at the end, which the headmaster accommodated to some extent. Building on the students’ ideas and in cooperation with the researchers and the teachers, the content of the intervention was constructed. After this, in phase 2, continuing to use an empowerment approach, the students, teachers and the researchers collaborated to identify important knowledge concerning the effects of AST and to write the relevant information in a useful format for the parents. The entire group decided on the main headlines of the parental information, which were; “Health benefits”, “Traffic safety”, “Environmental benefits” and “Cooperate to succeed”. The students worked in smaller groups to create the content related to one headline per group and presented their contribution to the whole group. Finally, a few students edited the content, which was subsequently sent home to the parents. In phase 3, a parenting meeting was held to utilize and discuss their perceived facilitators and barriers. In addition, all students and parents were invited to try cycling with winter tires, the students were offered a safety check of their bikes and free winter tires were provided by the research team. In phase 4, students were asked to use AST for 4 weeks and the project included gamification elements designed by the teachers to inspire internal motivation for engaging in healthy behavior [23]. For example, the students marked every active school transportation on a poster in the classroom and the measurement became a joint progress bar of class efforts. They also had a weekly assignment to solve on their way to school. For instance, this could be measuring the time and distance used in AST for each child and calculating the average speed when using a bike or walking. The information collected in assignments was used in daily lessons.

### 2.3. Data Collection

Data collection was inspired by photovoice, which followed the three steps described below to enable the students to be involved in the whole process [24].
Step 1: In line with a previous study [25], the aim of the project was discussed and formulated together with the students to make it meaningful for them. Apart from an explanation of the photovoice methodology and what it means to be a part of a research project, the introduction included a discussion about ethical issues regarding photographing and safety issues (i.e., not biking and taking a picture at the same time). After this, inspired by Belon et al. [19], we asked the students to take pictures to answer the question: What is it like to walk or cycle to school in the wintertime? To capture this, the participants were invited to take pictures for one winter month of their AST using their own mobile phones and to post them in a closed Facebook group for students, parents and researchers, or to e-mail them directly to one of the researchers. To reduce researchers’ influence on what was included, the students were given minimal direction on the photography. Acting as experts, the students chose what to photograph.Step 2: The photographing was followed by two workshops led by one of the researchers, where students analyzed the pictures that they had taken. In total, the students had taken 105 pictures in connection to the project. The workshops were held at the school during school hours and each workshop lasted ~40 min. (a)During the first workshop, to let all of the students individually express their thoughts, the participants were asked to select at least two of their own pictures and to write a text associated with them guided by the questions: “what does the picture show?”, “what feeling do you get when you see the picture?” and “what does the picture signify to you?”(b)During the second workshop, one week later, the students discussed their photos in peer groups of four to seven students with one of the researchers acting as a facilitator. The focus group sessions were audio recorded. The purpose was to create themes of knowledge and to allow the photos to stimulate the discussion. This workshop was mainly guided by an interview guide with questions, such as: “what does the picture signify to me/to us?” and “can we identify similarities among the pictures?”. Finally, participants were asked to pick three pictures that best represented the question [26]: How is it to walk or cycle to school in the wintertime?Step 3: at 5 weeks post-intervention, the parents completed an e-mail-distributed questionnaire with open questions concerning why they chose to participate and if they hesitated, their experience of the project, prerequisites to continuing with AST and if their child had continued using AST.

### 2.4. Data Analysis

Drawing on the methodology of photovoice, the students were a part of the first phase of the analysis as they were involved in the process of selecting pictures, thereby contextualizing and co-analyzing the data [24]. Moreover, the interpretations and discussions around the pictures were both written and oral and made individually and collectively, assuring that all students had the opportunity to express their thoughts. In the second phase of the analysis, the collected qualitative data from the focus group discussions were transcribed and together with the captions of the pictures and the text from the parents’ answers on open questions, this was analyzed with qualitative content analysis, which was inspired by Graneheim and Lundman [27]. The focus of the analysis concerning the students’ experiences was on their interpretations and discussions of the pictures rather than the actual pictures. A.-K.L. and S.R. actively participated in the following procedure: (1) the text was first read through several times to obtain a sense of the overall data; (2) the text was divided into meaning units guided by the aim of the study; (3) during the abstraction process, the meaning units were coded according to its content; (4) the codes were compared, contrasted and sorted into preliminary categories, with the authors striving to stay close to the text throughout this step; and (5) the categories were sorted into one main theme and three sub-themes. (6) After this, the transcriptions of the interviews were then read again to ensure that nothing had been left out and that their interpretations were correct. During the data analysis, we moved back and forward between the steps until we reached consensus between the authors. M.L. and A.E. critically reviewed the analysis to enhance the trustworthiness of the study. Moreover, quotations were used to further strengthen the study’s credibility.

## 3. Results

The results were formulated into one main theme while three subthemes and representative photos and quotations from the transcribed text were used. The quotes are labeled with a pseudonym, parent (P) or child (C) and given a participant number.

### 3.1. Involvement and Togetherness Unlocks Motivation

The students were invited to be co-creators of the project and in the beginning, the students were asked about what would make them more motivated to use AST during wintertime. The students emphasized that being heavily involved throughout the project sparked their motivation. One of their suggestions concerned rewards and all the students agreed that it was more fun to have a class reward than individual rewards although the more competitive students would have appreciated the creation of competition between classes or schools. A few boys had created their own competition, being those who biked most during the winter. This was motivating as long as you had a chance of winning. To further involve the students, they jointly created an information folder about the project for their parents and were the first persons to negotiate with their parents about being a part of the project. The parents expressed that the information created by their child was interesting to read and one parent wrote:*“Reading the information of what my child had accomplished made me impressed about their broad view of what AST means to health and environment, and it became a discussion around the dinner table that evening”*.Carl P17

Undertaking the project together with their classmates was also described as a very motivating factor for using AST and succeeding with the joint challenge despite the extreme cold or considerable amount of snow and other difficulties. For example, one child described it as teamwork as “*Doing this together in the class is like playing in a soccer team, everyone needs to contribute to achieving success*” Peter C3. The joint progress bar, where the students marked their active school transportations on a poster in the classroom, helped to motivate the students.

The progress bar showed that the attendance was high throughout the project (Table 3). However, there was relatively less progress on Fridays. According to the students, they sometimes had other engagements that day that prevented them from using active school transportation (e.g., requiring them to leave their bike at school over the weekend). 

Another part of togetherness that was highlighted was being able to walk or bike with friends, which this enhanced their motivation and made the trip more fun. Figure 1 and Figure 2 show pictures that exemplify the students’ discussion of togetherness.

During the project, the students offered each other social support. Furthermore, most students reported that their parents were supportive and acknowledged their performance. A few of the students had parents that cycled during wintertime and doing this together was inspiring and motivating. The students and parents had a Facebook site that was used for encouragement and to post pictures that show their experiences of using AST during wintertime. Most students thought that using social media to encourage each other to use AST was an easy and natural way to communicate. However, many noted that Instagram or Snapchat, which they typically used, would have been more convenient. Facebook was considered to be more suitable for parents. However, some noted that Instagram tends to not be serious while pictures cannot be saved in Snapchat. Many used Snapchat to tell their classmates that they were walking or biking that morning. The message had a positive effect on them as it was easier to join a friend and thus, this contributed to more of them using AST. Some parents had offered to drive their students when the weather circumstances were extremely bad but the students had denied the opportunity due to the project and their willingness to contribute to the success of it.

### 3.2. Parents Moving from Hesitant to Positive

Almost all of the students biked to school during fall and spring but only one child had previously used a bicycle during winter. Parents were identified by the students as the gatekeepers that permitted them to use AST during wintertime, with the parents’ approval being a key element determining the success of AST. Several parents expressed having doubts about letting their child bike when it was snowy and cold. The reasons for not letting their children cycle during wintertime included being close to the school, not having a bike and being concerned about safety. After the completion of the project, all of the parents of the children who cycled during winter were positive. One parent described the change in attitude as follows: *“My child was really eager to use winter cycling, but I just saw a picture in my head of her breaking her arms or leg, making me hesitant to this idea. Now I know that it is practically as safe as cycling during summer”* Elisabeth P9. Having winter tires was an important prerequisite before their children were allowed to try cycling during wintertime. The students expressed only a few doubts about using winter cycling and reported that having winter tires created a sense of safety. Furthermore, they mostly described it as being similar to biking on the snow-free ground. Nevertheless, biking with winter tires required attention on icy curves and knowing the path you are biking on made it easier to anticipate these areas. Some of the students said that their parents gradually became less anxious and some parents had borrowed their child’s bicycle and were surprised about how safe and easy it was cycling during wintertime. Parents were positive about not needing to drive their children to school and that their child arrived home earlier in the afternoon when they were able to cycle. Many parents said that this project had affected their attitude or own behavior towards more active transporting during wintertime. Some stated that it is hard to let the children bike in extreme cold and not do it oneself as: *”if it is possible for my child, then it is possible for me too”* Sarah P33. One parent described how it was easier to borrow their daughter’s bike to go and buy some milk than taking the car and planned to invest in winter tires for their own bikes after this project. Some had a long distance to travel to work but had become more positive about using active travel whenever possible. Figure 3 shows a picture exemplifying the students’ discussion about their parents´ attitudes towards AST.

### 3.3. Strenuous but Gives Instant Rewards and a Sense of Pride

Many of the students biking to school described being proud of their accomplishment of AST during these weeks. These students stated that it took commitment and grit to be able to take the bike every day despite the cold and snow and feeling tired in the mornings, which added to them not wanting to continue AST. Some students used positive thoughts, thinking of the feeling of satisfaction when arriving at school and the good feeling in their body after having exercised. Being able to accomplish this month of AST made them believe that they could accomplish many other things in life just by adopting the correct mindset and commitment.

The climate was not seen as an obstacle/barrier to using AST but rather as something they needed to relate to. For example, cycling in cold weather required being properly dressed for it even though covering the face was a challenge. The bikes functioned well if they were stored indoors during the night. Otherwise, the switches and lock easily froze, making it impossible to lock the bike at the schoolyard. However, the greatest difficulty with biking during wintertime was the snow, especially non-plowed roads. One of the most appreciated things with biking in wintertime was that it was a quick way of getting to and from school, which, for example, made it possible to sleep longer in the morning. All of the students thought that it was fun to bike, especially compared to walking to school. Many of the students also used the bike during their spare time and this elicited feelings of freedom as they were able to travel for longer distances without needing their parents to drive them. Walking or biking to school made the students wake up in the morning and feel more alert when the school day started. It was easier to concentrate on their schoolwork. In addition, the students reported feeling healthier and getting exercise from AST as well as having more friendly attitudes towards the environment. Some students acknowledged that they were in a better mood actively transporting to school. One student expressed it as follows: “*I am not a morning person, often I feel moody, but during this month when I every day have walked to school, I have been in a good spirit when arriving at school”* John C18. For the students walking and biking to school in wintertime, it became a habit and all of the students reported that they had continued with active school transport after completing the project. Figure 4 and Figure 5 shows pictures that represents examples of the students´ discussion about their efforts in going to school by bike in wintertime. However, they also felt satisfied doing it at the same time.

## 4. Discussion

We found that cycling to school during winter was highly feasible for students despite the extreme cold and snow. This is consistent with the findings of Mitra et al. [28], who found that children living within the sidewalk snowplow zone were less likely to walk to school than children living outside of the zone. Furthermore, they concluded that seasonality and short-term weather conditions do not appear to limit AST in general. Another aspect of AST is the impact of individual and social factors, which was suggested by Ducheyne et al. [29], as these two factors seem more important than the physical environment. Thus, interventions aiming to increase cycling should keep this in mind.

Being involved and doing the project together with their classmates was described as the most motivating factor to use AST. The intervention was created using the students’ ideas in cooperation with the researchers and the teachers. Involving students in the intervention development shed light on aspects that were different to those proposed by adult researchers and empowered children to influence their everyday life [20]. Furthermore, using photovoice appears to contribute to empowerment [15] and supporting participatory planning is essential in achieving the goals for interventions promoting physical activity [17,19]. This is important since previous research shows that empowerment enhances implementation and contributes to greater compatibility with user needs [30]. In turn, this increases the likelihood that programs will be sustained [31]. In addition, using the power of involving the whole class and including the parents in the intervention seems to have contributed to the projects’ success. Nevertheless, it could be argued that involving people as a group could diminish the individual’s sense of power as peer pressure makes it difficult for individuals to not follow the rest of the group [32]. Therefore, these aspects of power and empowerment need to be carefully considered when designing future interventions.

Our findings show that the students appreciated the sense of togetherness that the intervention created and the parents were supportive and acknowledged their children’s performance during the project. This is consistent with the social cognitive theory [21], which extends the conception of individuals to the collective as people do not operate as isolated individuals but work together to improve their health. Previous studies have also emphasized the importance of social support from peers and parents concerning health behaviors in general [33] and AST [34]. Our previous research regarding Pokémon GO (Niantic Inc., San Francisco, CA, US) also shows that cooperation conquers competition and this is consistent with the findings in this study as it was more motivating and fun to have a class reward than individual rewards [35]. In this study, Facebook was used to post photos and as a way for the participants to encourage each other to use AST. The participants reported that this was easy and a natural way of communicating, which is consistent with previous findings where Facebook has been assessed for physical activity interventions [36]. However, some of the students would have preferred to use other social media and spontaneously used Snapchat to motivate their classmates to use AST.

The students identified their parents as their gatekeepers that permitted them to use AST. Several parents expressed doubts and concerns with regards to their child cycling in winter, and after the intervention, when the parents had experienced how their children could cycle safely during winter, they altered their attitudes and became more positive. This finding is consistent with Bandura’s theory, which includes a belief that an individual’s ability to achieve a goal can be altered by direct mastery of experiences [21]. Even though parents’ attitudes are important and research shows that both the children’s and parent’s self-efficacy (i.e., parents’ belief in the child’s competence to safely use AST) affects the mode of transport [37], focusing on eliciting a change in attitudes could be questioned. Our previous research addressing AST during the snow-free period with 7-year-old children found that children’s use of AST can diminish parents’ perception of barriers and concerns and that it is likely more important to focus on increasing the use of AST than aiming to change their attitudes [38]. In addition, the conclusion was that it was beneficial to use the enthusiasm and motivation of children to overcome parental hesitation with AST. Kroesen et al. [39] found that aiming to change behavior was at least as effective as focusing on changing parental attitudes, which is supported by Festinger’s theory of cognitive dissonance. In this theory, conflicting attitudes and behaviors produce a feeling of discomfort that leads to an alteration of attitudes to reduce discomfort and restore balance [40]. The results show that this project also affected the parents’ attitude and behavior concerning their own mode of transportation, resulting in increased active transportation during wintertime. To our knowledge, this phenomenon has not been previously reported in relation to physical activity apart from in two of our previous studies [38,41]. However, in a study concerning children’s oral health, Tolvanen et al. [42] reported that health promotion directed to children also improved their parents’ behavior and they concluded that children should be the focus of future health promotion programs.

Our findings showed that because of the cold and snow, using AST during winter conditions was not easy and it foremost required commitment and grit. Grit is defined as perseverance and passion for long-term goals [43] and it is a characteristic that can change and develop through life in line with life experience [44]. According to Cosgrove et al. [45], physical activity is one way of developing grit as children learn to set goals and achieve them by overcoming adversities. Furthermore, our findings show that succeeding with this commitment made them believe that they could accomplish many other things in life by just adopting the correct mindset. However, it is not only the individual aspects that influence the amount of active travel. Cultural norms in a society, such as the belief that being active is not just a choice but a way of life, have been shown to be consistent among countries having the most active children and youth overall (Slovenia, Zimbabwe and Japan) [46]. Among these countries, Japan received the best grade for active transportation and has a highly established walking to school practice. Furthermore, it is expected that no primary school should be located more than 4 km from the student’s home [46]. Foley et al. [47] has shown that children using active transportation reported 11 min more in moderate to vigorous physical activity and 18 min less in screen time per day. They concluded that active travel is associated with health-promoting composition of time across multiple behavioral domains, which supports the public health case for active travel.

A limitation of this study is that there are no data on the amount of AST used by individual students before entering the project. Nevertheless, the knowledge generated in this study about how to involve end-users in intervention development and research as well as targeting a change in behavior instead of attitudes, might inform future studies in the area. 

## 5. Conclusions

The findings of our study might have implications for public health practices. Specifically, our findings suggest that interventions for increasing students’ AST in winter conditions should focus on the motivational aspects for both children and parents instead of the physical environment. Empowerment and including the end-users throughout the project are key contributors to success and makes it possible to design a sustainable program. This study also shows that targeting a change in behavior instead of focusing on changing attitudes could be beneficial in overcoming parental hesitation with AST during winter conditions. Furthermore, despite a small seasonal impact, using AST during winter is not that different from AST during fall and spring and therefore, interventions designed to facilitate AST could preferably target the season with most challenges as this would likely increase the proportion using AST all year around. 

## Figures and Tables

**Figure 1 ijerph-16-00234-f001:**
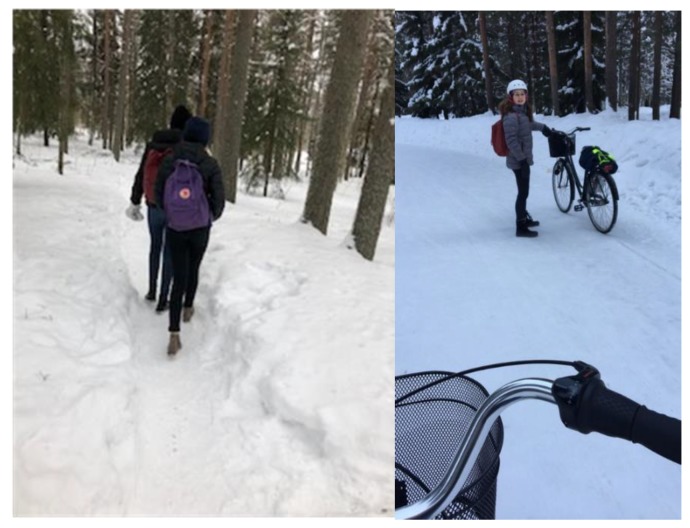
*“I have company on the way to school, and that is much more fun, often time flies when you walk or bike and talk. Even though it is cold outside, it is much warmer to walk together with friends”* Catrine C5.

**Figure 2 ijerph-16-00234-f002:**
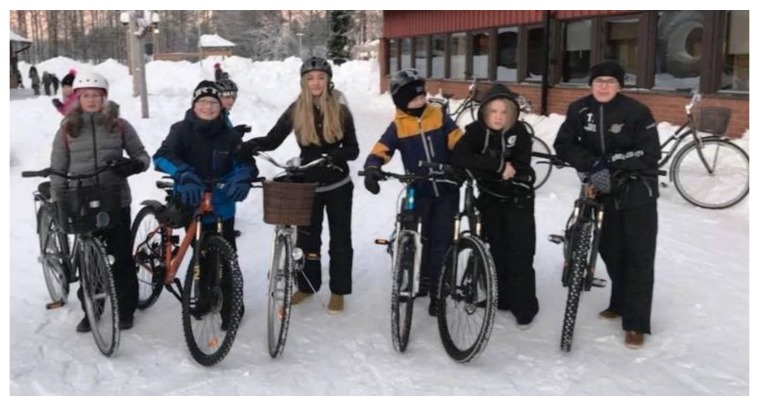
*“When I arrive at school, and I see a lot of friends with bicycles there I feel a fellowship with the others biking to school; it is like some kind of belonging”* Mathias C33.

**Figure 3 ijerph-16-00234-f003:**
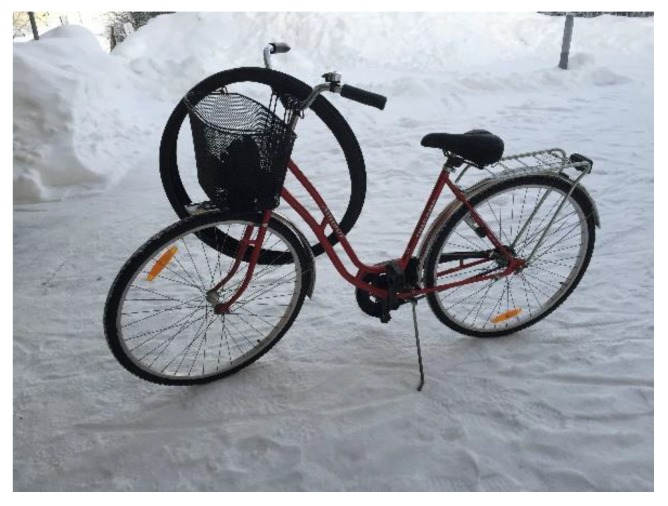
*“This is a picture of my bike the first day I got new tires; I remember that my parents were quite anxious, but then they tried the bike and got much more positive. I think that now after the project that they do not even think about me biking to school, it is nothing more special than doing it in the summertime”* Sofia C26.

**Figure 4 ijerph-16-00234-f004:**
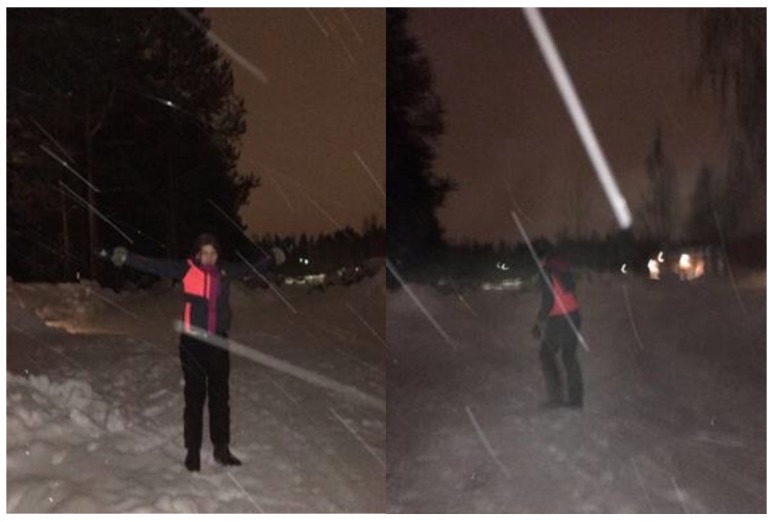
*“It is early in the morning and really cold outside, snowy and windy. Despite this I do not hesitate to walk to school, I get fresh air, and it is actually nice when I have accomplished it”* Ida C12.

**Figure 5 ijerph-16-00234-f005:**
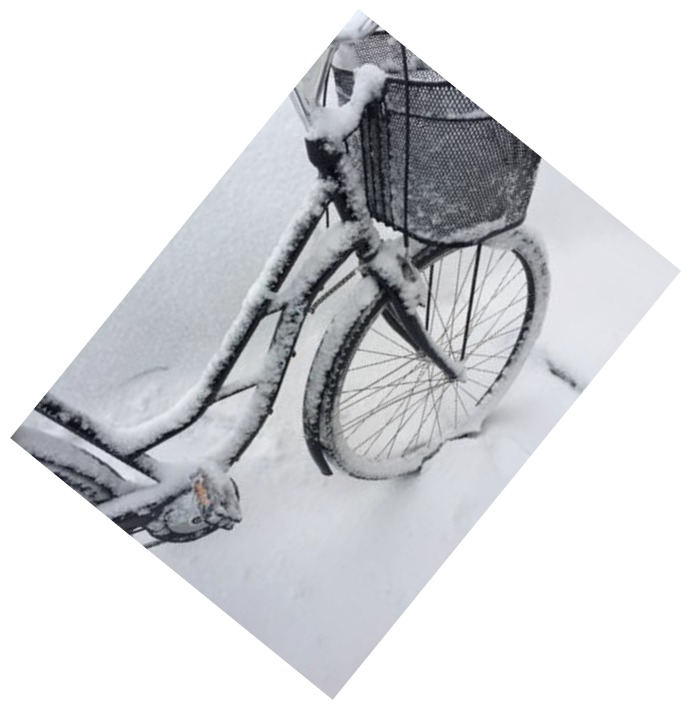
*“My bike has been outside all night, despite this I really want to take the bike to school; it is much faster than walking, more fun and you get exercise as well. The picture makes me proud and satisfied that I have accomplished going on this bike every day to school, even though I have 3 km one way, this way I have contributed to a better environment”* Josefine C22.

**Table 1 ijerph-16-00234-t001:** Characteristics of the municipality, schoolchildren and parents who participated in the study.

The context	This study was a substudy in a 3-year public health project concerning students’ active school transportation. The study was performed in a municipality with approximately 80,000 inhabitants situated in the northern part of Sweden.
The climate	The climate is characterized by harsh winters and sunny but short summers. AST^1^ during wintertime in northern Sweden means transportation on snow and ice and being outside in the extreme cold. During the first week of the project, it was between −23 °C to −31 °C. During the following weeks, the temperature varied between −15 °C to −20 °C (SMHI^2^, 2018). The sun rose between 08:15 h (February 5th) and 06:36 h (March 2nd) and the sunset was between 15:20 h and 16:48 h during this period.
The primary school	Two-hundred-seventy children attended the school, which was situated in a neighborhood with apartment buildings and detached houses. The distance to the school varied between 0.6 and 3.0 km, with an average of 1.3 km. The area contains both roads with sidewalks and roads without as well as different levels of safety at crossings.
The students	The two classes consisted of 40 students aged 12–13 years (20 boys and 20 girls). Thirty-five students (15 boys and 20 girls) agreed to participate in the study. Eighteen of them (9 boys and 9 girls) chose to cycle in winter and 17 (6 boys and 11 girls) chose to walk. Everyone (35) participated in the individual data collection (the first workshop). Thirty-three participated in the focus group discussions (the second workshop) since two boys were sick that day.
The parents	Thirty-four parents (22 women and 12 men) participated and they were between 39- and 56-years-old. Thirty of them had a university education and 31 were born in Sweden.

Notes: AST: active school transportation; SMHI: Swedish Meteorological and Hydrological Institute.

**Table 2 ijerph-16-00234-t002:** Overview of the timeline of the study.

November 2017	December 2017	January 2018	February 5th to March 2nd 2018	March 2018 2–3 Weeks after AST	April 2018 5 Weeks after AST
Phase 1	Phase 2	Phase 3	Phase 4	Phase 5	Phase 6
Empowerment—getting students involved	Increase students´ knowledge	Involving parents and increasing their knowledge	Performing AST for 4 weeks, Data collection step 1.	Data collection, step 2, with the students	Data collection with the parents
Workshop with students and teachers to create intervention	Workshop about effects of AST, including creating parental information	Discuss and utilize perceived facilitators and barriers to AST with parents. Try cycling in winter.	AST including gamification elements. Data collection using photovoice,	Two workshops concerning the pictures taken by the students, held one week apart	Open questions about AST in a questionnaire to the parents

**Table 3 ijerph-16-00234-t003:** A summary of the joint progress bar.

Active School Transportation	Monday	Tuesday	Wednesday	Thursday	Friday
Week 1	35	34	35	28	31
Week 2	35	35	30	32	29
Week 3	33	32	30	27	23
Week 4	31	30	31	30	25

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
