# Peer review of "Active School Transportation in Winter Conditions: Biking Together Is Warmer"

_ijerph, 2019, doi:10.3390/ijerph16020234_

Round 1
Reviewer 1 Report
The manuscript focuses on active school transportation by walking and cycling in Winter conditions, which is an understudied topic within a now expansive literature on children’s school transportation. More specifically, the paper reports findings from a participatory community intervention implemented in one school in Northern Sweden. A student- and parent-led qualitative study was designed focusing particularly on the photovoice approach, providing insights into the perceptions of children and parents relating to barriers and motivations.
I have a few concerns around the study method, which need to be clarified before the implication of this study can be fully understood. First, of the 40 students who participated in the study, were they all being driven to their school in winter months before this intervention? If some of them were already performing AST, then what do we learn about the impacts of this intervention?
Second, Some of the participatory children decided to walk while the rest choose to cycle. The barriers and motivations relating to these two different modes can be very different, but the discussion of results did not emphasize such differences.
Third, Table 4 indicates that participation in AST somewhat declined with time. That is, participation in weeks 3 and 4 were lower than in the first two weeks. Once could argue that as a result of such intervention, students find motivation for a while but this may not be a sustainable approach to encourage AST? The manuscript included no discussion on this observed trend.
My other comments are summarized below:
Abstract: Scope needs to be defined more clearly. There is a large literature now on the topic of active school transportation. So the "winter conditions" should be more clearly defined here and elsewhere.
Page 1, line 41: A large body of literature has been produced over the past decade discussing neighbourhood-level interventions.
Page 2: The project and the nature of intervention should be more clearly described at the beginning of the Methods section.
Page 2, line 65: Please discuss the Social Cognitive Theory and also clearly contextualize your work within this theoretical framework. Just a mention of a theoretical concept makes the manuscript unnecessarily confusing. Why is it appropriate to evaluate your intervention within this particular theoretical framework?
Table 1: What does “perform AST” mean? Were these students not walking ot biking to school before the intervention?
Table 2 is really useful and nicely summarizes the context of this work.
Page 4, line 125: “each data collection event took about 40 minutes.” Please clarify.
Page 4, line 133: Please replace “tape recorded” with “audio-recorded”.
Table 3: The study identifies the key theme as being “children’s winter cycling….is feasible and rewarding.” But surprisingly, this theme was never discussed. Please report results pertaining to this broad finding before discussing the sub-themes.
Page 5, lines 165-177: Much of this discussion focuses on “rewards” and not necessarily involvement or togetherness. Please review and revise.
Author Response
We are really grateful for your time and effort to improving our manuscript and appreciate your useful comments below.
The manuscript focuses on active school transportation by walking and cycling in Winter conditions, which is an understudied topic within a now expansive literature on children’s school transportation. More specifically, the paper reports findings from a participatory community intervention implemented in one school in Northern Sweden. A student- and parent-led qualitative study was designed focusing particularly on the photovoice approach, providing insights into the perceptions of children and parents relating to barriers and motivations.I have a few concerns around the study method, which need to be clarified before the implication of this study can be fully understood. First, of the 40 students who participated in the study, were they all being driven to their school in winter months before this intervention? If some of them were already performing AST, then what do we learn about the impacts of this intervention?
This is an important comment, and we have clarified in the result that only one child used winter bicycling prior to the project. However we lack further information about the amount of AST used by the children before the intervention and we have clarified in the discussion section that this is a limitation in this study.
Second, Some of the participatory children decided to walk while the rest choose to cycle. The barriers and motivations relating to these two different modes can be very different, but the discussion of results did not emphasize such differences.
Our approach in this study was to get the children use active transport and as a part of this we wanted to encourage them to try winter bicycling. The parents presented reasons for not choosing to let their child try winter bicycling, like living to close, not having a bike available or being hesitant about this due to risk of injuries. We are sure that this is not the whole truth, probably choosing what their friend chose and other reasons impacted the choice. As we have so little data on this subject we have chosen not to discuss it but this is absolutely an aspect that we will take in consideration in future studies.
Third, Table 4 indicates that participation in AST somewhat declined with time. That is, participation in weeks 3 and 4 were lower than in the first two weeks. One could argue that as a result of such intervention, students find motivation for a while but this may not be a sustainable approach to encourage AST? The manuscript included no discussion on this observed trend.
We discussed this with the teachers and they told us that the first two weeks of the project in total 2 children were sick part time of these weeks, however in week 3-4 it was 4 children that were sick, which may explain some of this trend. As the teachers was not included in the data collection we found it hard to describe this in the result and choose to not highlight it in the discussion part either. If you still find it vital to discuss this we will of course provide a discussion about it.
My other comments are summarized below:
Abstract: Scope needs to be defined more clearly. There is a large literature now on the topic of active school transportation. So the "winter conditions" should be more clearly defined here and elsewhere.
Thank you for this comment. We have clarified the scope in the abstract, being AST in winter conditions. We have as well tried to more clearly define the winter aspect of this study in the introduction.
Page 1, line 41: A large body of literature has been produced over the past decade discussing neighbourhood-level interventions.
We have added text and a reference describing that there is support for the benefits of community-based interventions, however, there are gaps for instance concerning school based programs.
Page 2: The project and the nature of intervention should be more clearly described at the beginning of the Methods section.
To improve this aspect we have clarified the content under the heading “Design”. We have as well rewritten the part under “procedure and participants” to make the procedure of the intervention more clear.
Page 2, line 65: Please discuss the Social Cognitive Theory and also clearly contextualize your work within this theoretical framework. Just a mention of a theoretical concept makes the manuscript unnecessarily confusing. Why is it appropriate to evaluate your intervention within this particular theoretical framework?
We appreciate this comment and we agree with you. Therefore, we have tried to briefly give a rational to why we used Social Cognitive Theory in the Design section.
Table 1: What does “perform AST” mean? Were these students not walking ot biking to school before the intervention?
We are grateful for this comment as it made us aware of the problem of understanding table 1. We have made a lot of clarifications in the table and also in the text connected to the table, which hopefully will make this more understandable. Unfortunately we do not have data on how much AST the students used before this intervention. We know that only one child used winter bicycling and both the students and the parents describe that the use of AST has increased.
Table 2 is really useful and nicely summarizes the context of this work.
Page 4, line 125: “each data collection event took about 40 minutes.” Please clarify.
We have changed the words “data collection event” to workshop instead to clarify this.
Page 4, line 133: Please replace “tape recorded” with “audio-recorded”.
We have corrected this.
Table 3: The study identifies the key theme as being “children’s winter cycling….is feasible and rewarding.” But surprisingly, this theme was never discussed. Please report results pertaining to this broad finding before discussing the sub-themes.
Thank you for this comment, to save words we earlier chose to not provide any text, however revising this made us aware of the strength of having clarified this abstraction level.
Page 5, lines 165-177: Much of this discussion focuses on “rewards” and not necessarily involvement or togetherness. Please review and revise.
We have rewritten this and hopefully it is more clear that it is about involvement and togetherness that unlocks motivation and not only the reward.
We believe that your help has really improved our manuscript and we are grateful for that.

Reviewer 2 Report
General comment:
I was very excited to read the article as the topic, methodology and methods used sounded very interesting and exciting. After reading the article I am still convinced that the paper should be published after some major revisions as there was at some points too much or too little information provided as well as often the presentation of findings were not backed up by the data (e.g. statements made without quotes supporting the statement). I also find that the presentation of findings and the discussion of findings do not match and therefore need to be aligned more explicitly.
Abstract: The result section needs to be rewritten as just presenting the themes presented in the article does not provide any information about the findings per se. So, what are the main findings rather than the themes that derived from the analysis?
Methods: what are open letters? What is the age of participants e.g. primary school children and location of the study
Introduction:
The introduction combined a general introduction to the topic and a very brief literature review and justification for the method chosen so that neither of the aims for this section could be done justice.
While there is limited research on AST during winter times, it is surprising that the authors do not turn to studies who look for example at play habits, barriers and norms during winter time e.g. (Ergler et al. 2013 and 2016, Lim , Hoordyk, 2015) to enrich both the literature review, but also their discussion later on.
The authors further might find it helpful to read Tim Ingolds, Phillip Vanini or xxx work for enriching both their discussion and the literature review.
I am surprised about the detailed, but superficial justification of photo-voice as a method given that it is a common method to elicit children’s experiences and views and the benefits for knowledge transfer are widely known. Rather I see the interesting aspect of the study in their participatory methodology and the researchers might find it helpful to review some studies on employing children as researchers as some of the data has been analysed by the children (e.g. Kellet, Porter, Ergler)
Moreover, a discussion about cultural norms, attitudes and policies are missed in the presentation of the literature e.g. cultural politics to create school zones which impact on AST behaviour in the US, outdoor/active travel attitudes of some countries etc. to provide more context for the study, but also to foreshadow in which direction the presentation of findings and discussion moves.
My suggestion is to extend the introduction/literature review substantively and focus on the most important aspects for the presentations of materials and methods
The introduction further remains unclear what age group the study works with and what the main argument of the article is
Materials and methods
The section on Design is very confusing as it foreshadows somehow what practical approach and methods have been used without going into enough detail. My suggestion is to rewrite this section and focus on the step the researchers undertook e.g. consultation/collaboration with the children, setting up intervention, etc. rather than a mix of methods, analysis.
Table 1 can be helpful when revised, but currently the information provided is not detailed enough to make sense of what was actually going on in each phase or the purpose of the phase eg later in method section the authors say that two workshops have been conducted with the children with the first one individually reflecting on children’s photos and the second one collaboratively reflecting and comparing experiences or e.g. Increase knowledge, but about what? For whom?
I am also surprised about the sentence that the authors conducted research previously in summer time, why is this necessary to know?
Procedure and participants
Not enough information about participants provided in the text e.g. age is missing in Table 2 and the text for children
Procedures could be more clearly structured e.g. discussing and presenting the different steps
How many more children who did not get parental consent would have loved to participate?
What type of editing has been done to the document and by whom? How was this negotiated with the children? Did the researchers add or edit the children’s drafts based on the requirements from the local ethic board?
Table 2, is informative, but what about the daylight hours during this time and how do they effect children’s travel, how does the geography and topography as well as walking/cycling infrastructure of the area look like and what impact does this have on the ease of traveling
More information in the intervention and gaming activities needed. So, by whom have these be designed , what activities where involved, what information did the children receive e.g. a daily reminder, app, how did the games look like, did a researcher come to class every week, …
Although I don’t find the information on wearing a helmet important for understanding the study design more context is needed as it seems helmets are not compulsory for children in Sweden
Data collection
The first part of the data collection reads more like a literature review which has not been undertaken for the other sub-sections and is an inconsistency, but also too much detail is provided e.g. on the ethical protocol for undertaking photo-voice.
Where the children limited on how many photos they take? Biking might cause some issues for actually taking photos (see Walker), so how has that been dealt with? How many pictures were taken in total?
More clarification is needed on what actually happened during the workshops, the purpose of each one and how the workshops were structured..why were children only allowed to select 2 photos? What happened to the other ones?
What time period had the children to do to the photo-voice activity, where they given a device or were they expected to use their own cellphone?
Had everyone of the participants access to the facebook pages or just the researchers? Why were parents allowed access to the facebook page? Had teachers access, too?
More information on the open letter is needed, was this a an open ended questionnaire for parents?
Presentation of coding and findings themes not really clear
So a more concise and focus presentation of the research design, methods and analysis is needed to free space for a more substantive literature review and integration of qualitative data into the finding section
Results
In general the findings are not backed up by the data per se. Parts are nicely presented, but in other paragraphs no quotes are used or integrated, which I quite surprising.
Table 3 does not add anything to the article and should be removed
Why was hot chocolate only served on Mondays? Or why are hot chocolates on Mondays so special?
How is bad weather defined and by whom? How did the children conceptualise bad weather?
Table 4, why haven’t the progress bar been divided into walking or cycling? Has there been a difference in uptake as for example more children walked as it is easier to be picked up after school rather than leaving your bike at school over the weekend
I am also surprised given the empowerment approach the authors have adopted that they don’t use pseudonyms for the participants or let them chose themselves, which might be worth to consider for future projects
Discussion
My suggestion is to split the discussion in a discussion about the findings per se situating the findings more firmly in the literature and a wider body of literature (especially on soial norms and attitudes, experiences and some of the discussion is actually not related to any of findings presented previously, but introduces new findings/aspects) and have a second section on lessons learned as the children clearly transformed the initial design of the project and move beyond…which has not been considered for informing future policies e.g. providing people with the opportunity to choose their own communication, preference for class rewards and competitions
The discussion about UNCRC and Shier’s model seems redundant as it appears suddenly and does not add anything at this stage, it rather celebrates the study uncritically and given the discussions on employing children/young people as researchers that no issues arose is highly doubtful
There is no discussion about children’s individual experiences, what they liked/felt difficult
So, a clearer focus/argument is needed not only for the discussion but entire article
Author Response
General comment:
I was very excited to read the article as the topic, methodology and methods used sounded very interesting and exciting. After reading the article I am still convinced that the paper should be published after some major revisions as there was at some points too much or too little information provided as well as often the presentation of findings were not backed up by the data (e.g. statements made without quotes supporting the statement). I also find that the presentation of findings and the discussion of findings do not match and therefore need to be aligned more explicitly.
We are really grateful for your time and effort to improving our manuscript and appreciate your useful comments. In total we have rewritten large parts of the manuscript and you can see our specific answers below.
Abstract: The result section needs to be rewritten as just presenting the themes presented in the article does not provide any information about the findings per se. So, what are the main findings rather than the themes that derived from the analysis?
We have rewritten the result section in the abstract, and appreciate the comment as it made this part more interesting.
Methods: what are open letters? What is the age of participants e.g. primary school children and location of the study
We have changed “open letters” to “open questions in a questionnaire” throughout the manuscript to make this clear.
Introduction:
The introduction combined a general introduction to the topic and a very brief literature review and justification for the method chosen so that neither of the aims for this section could be done justice.
We have rewritten most of the introduction to deepen the literature review and justify the aim more, and as well making the winter aspects more clear.
While there is limited research on AST during winter times, it is surprising that the authors do not turn to studies who look for example at play habits, barriers and norms during winter time e.g. (Ergler et al. 2013 and 2016, Lim , Hoordyk, 2015) to enrich both the literature review, but also their discussion later on.
Thank you for these literature suggestions we have chosen to use one of the articles Ergler et al. 2013.
I am surprised about the detailed, but superficial justification of photo-voice as a method given that it is a common method to elicit children’s experiences and views and the benefits for knowledge transfer are widely known. Rather I see the interesting aspect of the study in their participatory methodology and the researchers might find it helpful to review some studies on employing children as researchers as some of the data has been analysed by the children (e.g. Kellet, Porter, Ergler)
We agree with you that Photovoice is a common method to elicit children´s experience and well known at least among researcher who is used to qualitative method. However, a lot of our colleagues are not so familiar with this method and probably also some of the readers of this journal so therefore, we would like to keep information about the method in the introduction. However we have rewritten the design section hopefully making the information more balanced. We have complemented the aspect of employing children in research with information from “Beyond Passive Participation: Children as Collaborators in Understanding Neighbourhood Experience. Ergler (2011)” who describes this in an excellent manner. Thank you for this suggestion.
Moreover, a discussion about cultural norms, attitudes and policies are missed in the presentation of the literature e.g. cultural politics to create school zones which impact on AST behaviour in the US, outdoor/active travel attitudes of some countries etc. to provide more context for the study, but also to foreshadow in which direction the presentation of findings and discussion moves.
We have tried to complement the introduction with these aspects.
My suggestion is to extend the introduction/literature review substantively and focus on the most important aspects for the presentations of materials and methods
We have followed your suggestion and extended the introduction and shortened the method section as well as rewritten parts to make it more clear.
The introduction further remains unclear what age group the study works with
We have clarified in the abstract the age of the children and this is also written in table 1.
Materials and methods
The section on Design is very confusing as it foreshadows somehow what practical approach and methods have been used without going into enough detail. My suggestion is to rewrite this section and focus on the step the researchers undertook e.g. consultation/collaboration with the children, setting up intervention, etc. rather than a mix of methods, analysis.
We have rewritten the “Design” section and focus on the intervention instead.
Table 1 can be helpful when revised, but currently the information provided is not detailed enough to make sense of what was actually going on in each phase or the purpose of the phase eg later in method section the authors say that two workshops have been conducted with the children with the first one individually reflecting on children’s photos and the second one collaboratively reflecting and comparing experiences or e.g. Increase knowledge, but about what? For whom?
We are grateful for this comment as it made us aware of the problem of understanding table 1 (which now is labeled Table 2). We have made a lot of clarifications in the table and also in the text connected to the table, which hopefully will make this more understandable.
I am also surprised about the sentence that the authors conducted research previously in summer time, why is this necessary to know?
We have deleted this sentence.
Procedure and participants
Not enough information about participants provided in the text e.g. age is missing in Table 2 and the text for children
To save word we have chosen to present most of the data of the participants in the Table were age of the children is on the first row. We have as well complemented with text about daylight and as well how the neighborhood was built.
Procedures could be more clearly structured e.g. discussing and presenting the different steps
We have clarified the steps of the procedure and explained it as phases 1-6. We have also rewritten a lot of the text to make this more understandable.
How many more children who did not get parental consent would have loved to participate?
Interesting question, however we have not collected data on this aspect.
What type of editing has been done to the document and by whom? How was this negotiated with the children? Did the researchers add or edit the children’s drafts based on the requirements from the local ethic board?
The children was involved in the design of the intervention and as well negotiating with their parents. They took the photos and their experiences that was displayed in the result. The information that they wrote to the parents was produced and edit by the children, but the workshop was organized by the teachers and researcher. The data analysis was done by the researchers and in retrospect, we can acknowledge the advantage of having children involved in this stage as well, however, this was not done in this study.
Table 2, is informative, but what about the daylight hours during this time and how do they effect children’s travel, how does the geography and topography as well as walking/cycling infrastructure of the area look like and what impact does this have on the ease of traveling
We have complemented the information with time when the sun rose respectively set. In total, the time of daylight increased with about 3 hours during this period. As the children´s home was located up to 3 km from the school the infrastructure differed quite a lot from each other it is hard to describe the area correct but we have added information about infrastructure more generally.
More information in the intervention and gaming activities needed. So, by whom have these be designed , what activities where involved, what information did the children receive e.g. a daily reminder, app, how did the games look like, did a researcher come to class every week, …
We have added information about the gamification elements of the intervention, which was designed by the teachers. No digital devices was used in the gamification part. The researchers did not attend the school at all during the period of AST, only before and after this period.
Although I don’t find the information on wearing a helmet important for understanding the study design more context is needed as it seems helmets are not compulsory for children in Sweden
Helmets are compulsory in Sweden, however, that does not guarantee that they are worn. However, we have removed the information about the helmets since it is more confusing then contributing to the text.
Data collection
The first part of the data collection reads more like a literature review which has not been undertaken for the other sub-sections and is an inconsistency, but also too much detail is provided e.g. on the ethical protocol for undertaking photo-voice.
We have rewritten the section Data Collection and structured it in Step 1-3 to hopefully making it more consistent.
Where the children limited on how many photos they take? Biking might cause some issues for actually taking photos (see Walker), so how has that been dealt with? How many pictures were taken in total?
The children were allowed to take as many picture as they liked. We have clarified that we talked to them about safety issues, like taking picture while biking. We have also added information about how many pictures that was taken in connection with the project and either posted on facebook, or sent to us through e-mail.
More clarification is needed on what actually happened during the workshops, the purpose of each one and how the workshops were structured. why were children only allowed to select 2 photos? What happened to the other ones?
We have clarified the purpose of the workshops and that they were asked to select at least 2 photos. The photos that was not chosen by the students was not brought into the discussion with them during the second workshop.
What time period had the children to do to the photo-voice activity, where they given a device or were they expected to use their own cellphone?
They took photos during the whole period of AST (4 weeks). They used their own device and we have added this in the text.
Had everyone of the participants access to the facebook pages or just the researchers? Why were parents allowed access to the facebook page? Had teachers access, too?
We have clarified that children, parents and the researcher had access to the closed facebook group. Teacher had not access.
More information on the open letter is needed, was this a an open ended questionnaire for parents?
We have clarified and changed “open letter” to “open questions in a questionnaire” throughout the manuscript.
Presentation of coding and findings themes not really clear
We have rewritten parts of the Data analysis, to enhance the understanding of the process.
So a more concise and focus presentation of the research design, methods and analysis is needed to free space for a more substantive literature review and integration of qualitative data into the finding section
Results
In general the findings are not backed up by the data per se. Parts are nicely presented, but in other paragraphs no quotes are used or integrated, which I quite surprising.
We are grateful for this comment as it give us the opportunity to include more quotes. We have added 2 more quotes and one picture with a quote.
Table 3 does not add anything to the article and should be removed
We have removed Table 3.
Why was hot chocolate only served on Mondays? Or why are hot chocolates on Mondays so special?
We have removed the text about serving hot chocolate on Mondays, as it could be percieved as irrelevant.
How is bad weather defined and by whom? How did the children conceptualise bad weather?
We have changed “bad weather” and explained that the children or parents were referring to.
Table 4, why haven’t the progress bar been divided into walking or cycling? Has there been a difference in uptake as for example more children walked as it is easier to be picked up after school rather than leaving your bike at school over the weekend
We did not ask the teachers to have different progress bars for the class, as this was seen as a joint class project. However, we can see that it would have been an advantage to have that data, especially as some of the children explained that the reason for not using AST on Fridays was having other engagements that day, and that for example, leaving their bike on school over the weekend was a problem.
I am also surprised given the empowerment approach the authors have adopted that they don’t use pseudonyms for the participants or let them chose themselves, which might be worth to consider for future projects
Thank you for this suggestion, we have now added pseudonyms in the quotes.
Discussion
My suggestion is to split the discussion in a discussion about the findings per se situating the findings more firmly in the literature and a wider body of literature (especially on soial norms and attitudes, experiences and some of the discussion is actually not related to any of findings presented previously, but introduces new findings/aspects) and have a second section on lessons learned as the children clearly transformed the initial design of the project and move beyond…which has not been considered for informing future policies e.g. providing people with the opportunity to choose their own communication, preference for class rewards and competitions
Thank you for making us take the whole Discussion under consideration and giving us the chance to improve it. We have rewritten and restructured large parts. We have also deleted some discussion that was not related to findings, such as that the finding suggest that photovoice can be a valuable method.
The discussion about UNCRC and Shier’s model seems redundant as it appears suddenly and does not add anything at this stage, it rather celebrates the study uncritically and given the discussions on employing children/young people as researchers that no issues arose is highly doubtful
We have deleted these sentences.
There is no discussion about children’s individual experiences, what they liked/felt difficult
We have in this study analysed all focusgroups interviews, the text from the pictures and the text from the open questions to the parents as a whole. The data was analysed and sorted into themes and than represented as a whole with all the different experience of each child merged in the finding. The quotes give example of individual experience and is used for the reader to verify the interpretation of the result. Therefore it is quite problematic to have a discussion about the children´s individual experience, even though we can see the value of doing that in the next study.
So, a clearer focus/argument is needed not only for the discussion but entire article
We believe that your help has really improved our manuscript and we are grateful for that.

Reviewer 3 Report
This paper reports on the implementation and outcome of a school based active transport project. The method of engaging students, student participation in the project and the results are clearly described. It is an engaging paper and offers a model that could be taken up and/or adapted to other locations.There are a few issues the paper needs to address.
· First, a minor issue. The paper demonstrates the efficacy of getting kids to bike to school but was there a shift from one mode of active travel (walking) to another (biking) or from passive (car based) to active travel (biking).
· Second, a major issue is the question of power and empowerment. The concept of empowerment and its links to broader governmental programs have been well critiqued (e.g. Barbara Cruikshank’s 1999 volume The Will to Empower). The current paper demonstrates an effective way of getting children to manage and align their behaviour with those of authorities (in this case the health researchers and, no doubt, government departments of health).
I have no issue with health researchers or government departments fostering healthy behaviour in children (or adults) in the way the researchers describe. Providing an alternative discourse about transport (as important to health) is excellent as it offers another way for children to think about how they travel. However, it seems disingenuous to describe empowerment as self-efficacy or exercising control over one’s life.
Consider the process of ‘empowerment’ described by the authors. Some researchers from the University (authoritative people from an authoritative institution) turn up to talk about joining in a program to bike to school. These researchers have the support of other authority figures (the teachers and principal). Did the children really stand a chance of saying ‘no’ to being part of ‘co-producing’ the program and ‘no’ to biking to school? It would take a kid with a lot of self-efficacy to say ‘no’. As the authors themselves note in their discussion ‘using the power of involving the whole class…[and] parents …contributed to the project’s success’. The literal and metaphorical gaze of the class community is a none-to-subtle pressure for each child to bring themselves into line and meet the requirement to participate and the standards of healthy behaviour that inform biking to school. How does that sit with self-efficacy? The gaze is also a well-used technique for bringing parents into line. Further, the fact that the students wanted a reward for their behaviour suggests to me they are thoroughly immersed in the disciplinary operation of power and the self-regulating, governable individuals it seeks to produce.
I have a lot of sympathy with this paper especially the view that fostering cooperation can have far more positive effects than competition. While I don’t expect the researchers to agree with my arguments on power, they do need to explain their understanding of empowerment and provide a far more nuanced and critical discussion of theories of power and empowerment.
Author Response
We are really grateful for your time and effort to improving our manuscript and appreciate your comments about power and empowerment.
This paper reports on the implementation and outcome of a school based active transport project. The method of engaging students, student participation in the project and the results are clearly described. It is an engaging paper and offers a model that could be taken up and/or adapted to other locations.
There are a few issues the paper needs to address.
· First, a minor issue. The paper demonstrates the efficacy of getting kids to bike to school but was there a shift from one mode of active travel (walking) to another (biking) or from passive (car based) to active travel (biking).
This is an important comment, and we have clarified in the result that only one child used winter bicycling prior to the project. However we lack further information about the amount of AST used by the children before the intervention and we have clarified in the discussion section that this is a limitation in this study.
· Second, a major issue is the question of power and empowerment. The concept of empowerment and its links to broader governmental programs have been well critiqued (e.g. Barbara Cruikshank’s 1999 volume The Will to Empower). The current paper demonstrates an effective way of getting children to manage and align their behaviour with those of authorities (in this case the health researchers and, no doubt, government departments of health). I have no issue with health researchers or government departments fostering healthy behaviour in children (or adults) in the way the researchers describe. Providing an alternative discourse about transport (as important to health) is excellent as it offers another way for children to think about how they travel. However, it seems disingenuous to describe empowerment as self-efficacy or exercising control over one’s life. Consider the process of ‘empowerment’ described by the authors. Some researchers from the University (authoritative people from an authoritative institution) turn up to talk about joining in a program to bike to school. These researchers have the support of other authority figures (the teachers and principal). Did the children really stand a chance of saying ‘no’ to being part of ‘co-producing’ the program and ‘no’ to biking to school? It would take a kid with a lot of self-efficacy to say ‘no’. As the authors themselves note in their discussion ‘using the power of involving the whole class…[and] parents …contributed to the project’s success’. The literal and metaphorical gaze of the class community is a none-to-subtle pressure for each child to bring themselves into line and meet the requirement to participate and the standards of healthy behaviour that inform biking to school. How does that sit with self-efficacy? The gaze is also a well-used technique for bringing parents into line. Further, the fact that the students wanted a reward for their behaviour suggests to me they are thoroughly immersed in the disciplinary operation of power and the self-regulating, governable individuals it seeks to produce.
We appreciate this valuable consideration on power and empowerment. We have added text in the Discussion part to lift this perspective, as it is important not to take for granted that involving the whole group in a joint goal is the best thing for everybody.
I have a lot of sympathy with this paper especially the view that fostering cooperation can have far more positive effects than competition. While I don’t expect the researchers to agree with my arguments on power, they do need to explain their understanding of empowerment and provide a far more nuanced and critical discussion of theories of power and empowerment.
In the Discussion part, we have discussed more about the involvement of the children and thereby some parts of empowerment as well as added the text mentioned above.
Round 2
Reviewer 1 Report
The revised manuscript has addressed many of my concerns. However, there are still areas for further improvement, which I have summarized below:
Please define AST in your introduction.
The Social Cognitive Theory is described in the revised manuscript, but there is still a need to contextualize this research within the scope of this theory. How does this theory help with the conceptualizations presented in this manuscript? Please provide more specific descriptions.
In response to one of my previous comments, the authors responded that "Our approach in this study was to get the children use active transport and as a part of this we wanted to encourage them to try winter bicycling". Please include this discussion in your study design.
Discussion relating to the theme "children’s winter cycling….is feasible and rewarding." is still inadequate and appears incomplete. Please discuss this section with as much importance as you have placed on other sub-themes, by using quotes etc. Otherwise, remove this.
Author Response
Thank you again for your valuable help to further improve this manuscript. We have sent the manuscript to an English editing company for a second time, to assure that the English is correct. These changes is not visible in the document.
The revised manuscript has addressed many of my concerns. However, there are still areas for further improvement, which I have summarized below:
Please define AST in your introduction.
We have defined AST as walking or biking to school.
The Social Cognitive Theory is described in the revised manuscript, but there is still a need to contextualize this research within the scope of this theory. How does this theory help with the conceptualizations presented in this manuscript? Please provide more specific descriptions.
We have added text and clarified how the Social Cognitive Theory supported the conceptualizations of this study.
In response to one of my previous comments, the authors responded that "Our approach in this study was to get the children use active transport and as a part of this we wanted to encourage them to try winter bicycling". Please include this discussion in your study design.
We have added text in the design part, to clarify this.
Discussion relating to the theme "children’s winter cycling….is feasible and rewarding." is still inadequate and appears incomplete. Please discuss this section with as much importance as you have placed on other sub-themes, by using quotes etc. Otherwise, remove this.
We have removed the theme.

Reviewer 2 Report
Thank you for sending your revised manuscript. I am a bit surprised about the still short literature review, but with the revisions undertaken in the methodology section the article seems more balanced now. However, the photovoice section still appears abrupt and no link has been made to the previous discussions.
So, my suggestion is to provide a better justification for the placement of this discussion in the literature review e.g. a lack of interventions looking at children/young people’s voices directly to understand their needs and experiences. At the moment, the link to photo voice still remains unclear.
A few minor additional suggestions
I further suggest to change your terminology from children to adolescents/teenagers as you conducted your study with young people rather than children or at least mention teenagers/adolescents in a footnote as this will enhance your citation given that quite a few researchers work with adolescents.
To add the school hours for Sweden in table 1 to get a sense whether the participants travel in the dark or daylight….out of curiosity…did any of the participants talk about the visability/daylight hours/time of the day when traveling?
Who did the editing process for the information that was sent home? Or where two documents sent home? One created by the young people on general information and then the required information sheets and consent forms?
Where the captions the children/young people wrote also analysed as the analysis section only talks about the transcripts from the focus group discussions?
Author Response
Thank you again for your valuable help of improving this manuscript.
We have sent the manuscript to an English editing company for a second time, to assure that the English is correct. These changes is not visible in the document.
Thank you for sending your revised manuscript. I am a bit surprised about the still short literature review, but with the revisions undertaken in the methodology section the article seems more balanced now. However, the photovoice section still appears abrupt and no link has been made to the previous discussions. So, my suggestion is to provide a better justification for the placement of this discussion in the literature review e.g. a lack of interventions looking at children/young people’s voices directly to understand their needs and experiences. At the moment, the link to photo voice still remains unclear.
We have clarified the justification for using photovoice in this study.
A few minor additional suggestions
I further suggest to change your terminology from children to adolescents/teenagers as you conducted your study with young people rather than children or at least mention teenagers/adolescents in a footnote as this will enhance your citation given that quite a few researchers work with adolescents.
Thank you for this comment, we have changed the word children to students instead in most places. Since most of them were 12 and a few recently turned 13 it was hard to choose between children and teenager. We have not highlighted these changes as it was more than 80 of them.
To add the school hours for Sweden in table 1 to get a sense whether the participants travel in the dark or daylight….out of curiosity…did any of the participants talk about the visability/daylight hours/time of the day when traveling?
The participants did not talk about the dark, an explanation of this could be that they are so used to it being dark in the wintertime, and as you see in the table the light during this month changed a lot, probably making this an even smaller issue for them.
Who did the editing process for the information that was sent home? Or where two documents sent home? One created by the young people on general information and then the required information sheets and consent forms?
We have clarified that the students was the ones who did the editing process of the information.
Where the captions the children/young people wrote also analysed as the analysis section only talks about the transcripts from the focus group discussions?
We have clarified that the captions also was a part of the analysed data.
